# The utility of diagnostic selective nerve root blocks in the management of patients with lumbar radiculopathy: a systematic review

Rebecca Beynon,[1] Martha Maria Christine Elwenspoek,[1,2] Athena Sheppard,[1,2] John Nicholas Higgins,[3] Angelos G Kolias,[4] Rodney J Laing,[4] Penny Whiting,[1,2] William Hollingworth[1]

For numbered affiliations see end of article.

**Correspondence to**
Dr Martha Maria Christine Elwenspoek; martha.elwenspoek@bristol.ac.uk

## ABSTRACT

**Objective** Lumbar radiculopathy (LR) often manifests as pain in the lower back radiating into one leg (sciatica). Unsuccessful back surgery is associated with significant healthcare costs and risks to patients. This review aims to examine the diagnostic accuracy of selective nerve root blocks (SNRBs) to identify patients most likely to benefit from lumbar decompression surgery.

**Design** Systematic review of diagnostic test accuracy studies.

**Eligibility criteria** Primary research articles using a patient population with low back pain and symptoms in the leg, SNRB administered under radiological guidance as index test, and any reported reference standard for the diagnosis of LR.

**Information sources** MEDLINE (Ovid), MEDLINE In-Process & Other Non-Indexed Citations, EMBASE, Science Citation Index, Biosis, LILACS, Dissertation abstracts and National Technical Information Service from inception to 2018.

**Methods** Risk of bias and applicability was assessed using the QUADAS-2 tool. We performed random-effects logistic regression to meta-analyse studies grouped by reference standard.

**Results** 6 studies (341 patients) were included in this review. All studies were judged at high risk of bias. There was substantial heterogeneity across studies in sensitivity (range 57%–100%) and specificity (10%–86%) estimates. Four studies were diagnostic cohort studies that used either intraoperative findings during surgery (pooled sensitivity: 93.5% [95% CI 84.0 to 97.6]; specificity: 50.0% [16.8 to 83.2]) or 'outcome following surgery' as the reference standard (pooled sensitivity: 90.9% [83.1 to 95.3]; specificity 22.0% [7.4 to 49.9]). Two studies had a within-patient case-control study design, but results were not pooled because different types of control injections were used.

**Conclusions** We found limited evidence which was of low methodological quality indicating that the diagnostic accuracy of SNRB is uncertain and that specificity in particular may be low. SNRB is a safe test with a low risk of clinically significant complications, but it remains unclear whether the additional diagnostic information it provides justifies the cost of the test.

## INTRODUCTION

In Western Europe, low back pain is the leading cause of disability and represents a high economic burden,[1] in particular due to production losses and cost of informal care.[1] In a subgroup of patients, low back pain is accompanied by pain radiating to a lower extremity in a radicular distribution (sciatic pain). Leg pain is one of the symptoms of lumbar radiculopathy (LR) but other symptoms, such as numbness, tingling, weakness, can also develop. LR can be the result of compressive or inflammatory disorders of the spinal nerve roots or a combination of these. Randomised trial evidence on the effectiveness of lumbar decompressive surgery in patients with radiculopathy and intervertebral disc herniation suggests that early surgery leads to faster pain relief, but longer-term effectiveness is less clear.[2–7] Current UK guidelines recommend spinal decompression

surgery for patients with radicular pain when non-surgical treatments have not improved symptoms and radiological findings are consistent with physical examination.[8] However, surgery does not always resolve radicular pain and 5%–36% of patients suffer from recurrent back and leg pain within 2 years postsurgery.[9] The main cause of unsuccessful back surgery is inaccurate diagnosis.[10] Improved diagnosis could help identify patients most likely to benefit from surgery and minimise the cost and risks associated with unsuccessful back surgery.

A timely and accurate diagnosis of the cause of low back pain and radicular pain is important, since it is occasionally an early symptom of serious systemic disease,[11] and an inaccurate diagnosis can lead to a cascade of costly, invasive and ineffective therapy. In most patients, the diagnosis of radiculopathy, caused by nerve root compression, is made by correlation of symptoms, clinical signs and imaging findings. However, neither clinical findings nor radiological imaging have perfect diagnostic accuracy.[12] When clinical and imaging findings are equivocal or discordant, uncertainty remains about the source of the symptoms and whether nerve root decompression will relieve symptoms. Additional diagnostic tests could help clinicians and patients to choose between surgical and conservative care or guide surgery in patients with suspected multilevel radiculopathy.

Diagnostic selective nerve root blocks (SNRBs) inject local anaesthetic or other substances around spinal nerves under imaging guidance. Both provocative responses (replicating symptoms during needle placement) and analgesic responses (significant reduction of symptoms) to SNRB may be diagnostically useful in confirming or ruling out a given nerve root as the source of clinical symptoms. Some clinical guidelines and consensus statements have endorsed the use of SNRB to identify the source of pain in patients with multilevel pathology and in the preoperative evaluation of patients with a negative or inconclusive imaging study.[13 14] Over the last decade, several systematic reviews have investigated SNRB as diagnostic tool, covering the literature up to 2012.[15–18] However, evidence was scarce and of low quality and the diagnostic accuracy and reliability of SNRB remained unclear. We updated our previous systematic review to determine the diagnostic performance of SNRB in addition to clinical and imaging findings for identifying patients with LR who are good candidates for lumbar decompression surgery.[15] A secondary aim was to summarise evidence on the incidence of procedure-related complications.

## MATERIALS AND METHODS
### Literature search
We updated the search from our previous review, searching all databases to March 2018. Our previous search aimed to identify published and unpublished studies by searching MEDLINE (Ovid), MEDLINE In-Process & Other Non-Indexed Citations, EMBASE, Science Citation Index, Biosis and LILACS (Latin American and Caribbean literature database), Dissertation abstracts and National Technical Information Servicefrom inception to March 2018. Our search strategy combined terms for SNRB with terms for sciatica or radiculopathy (see online supplementary search strategy).[15] We did not use a methodological search filter to identify diagnostic accuracy studies as such filters result in the omission of relevant studies.[19–21] No language restrictions were applied. Attempts were made to identify further studies by examining the reference lists of all included articles.

### Study selection
Studies were eligible for the diagnostic accuracy review if they included patients with low back pain and leg pain who underwent SNRB under imaging guidance. The studies needed to report sufficient data to construct a table detailing diagnostic accuracy (ie, numbers of true negative, true positive, false positive and false negative results) of the index test (SNRB) compared with any reported 'reference standard'. When we were unable to extract sufficient details from otherwise eligible studies we contacted study authors.

In diagnostic accuracy studies, the reference standard is typically a definitive test used to determine the true diagnosis, but no such definitive test exists for radicular pain due to nerve root compression. Therefore, most diagnostic studies used either intraoperative findings or postsurgical follow-up as the reference standard to judge the diagnostic accuracy of SNRB. An alternative approach is to determine the sensitivity of SNRB using a 'case' injection at a symptomatic nerve root level where nerve root compression is confirmed by imaging. Specificity is evaluated by a 'control' injection at an asymptomatic site (eg, adjacent nerve root) where imaging demonstrates no nerve root compression. Hence, in this approach, concordant clinical and imaging findings are used as the reference standard.

Two reviewers independently screened titles and abstracts for relevance and full papers for eligibility. Any disagreements were resolved by consensus or referred to the review team.

### Data extraction and quality (bias and applicability) assessment
Data extraction was performed by one reviewer and checked by a second: disagreements were resolved by consensus or discussion among coauthors. We extracted data on: study design, inclusion and exclusion criteria, included patients, SNRB details and reference standard details. 'Per patient' data were extracted: if these were unavailable we extracted 'per injection' data.

Studies included in the diagnostic review were assessed for methodological quality using the QUADAS-2 measure of bias and applicability.[16] Bias occurs if the results of a study are distorted by flaws or limitations in its design or conduct (eg, knowledge of the index test result when interpreting the reference standard). Applicability may be reduced if patient characteristics, or the use or

interpretation of the index test in the study differ from those likely to prevail in clinical practice. Reviewers rate concerns regarding applicability and risk of bias as low, high or unclear. At least two reviewers assessed quality using QUADAS-2 and any disagreements were resolved by consensus.[22]

Studies were judged to be of high applicability if: (1) they recruited patients with low back pain and suspected radiculopathy (sciatica) with non-congruent imaging and clinical findings, who might benefit from lumbar decompression surgery; (2) the SNRB included injection of anaesthetic, sometimes in conjunction with a steroid, close to the lumbar nerve root most often under guidance by fluoroscopy or other imaging; (3) the test aimed to identify patients with radiculopathy (sciatica) that was amenable to surgery and (4) the reference standard was outcome of surgery. We did not carry out formal quality assessment of studies reporting on adverse events.

### Data synthesis and analysis

We performed all analyses in Stata V.15.1.[23] We calculated sensitivity and specificity of SNRB from each study and plotted these in receiver-operating characteristic space. We performed random-effects logistic regression to meta-analyse studies grouped by reference standard,[24] using an updated version of the *metandi* package.[25] Data from studies on adverse events were combined in a narrative summary. We reported our findings according to the Preferred Reporting Items for Systematic Reviews and Meta-Analysesfor diagnostic test accuracy studies.[26]

### Patient and public involvement

Patients and members of the public were not involved in this review.

### RESULTS

The original searches identified 12 883 titles and abstracts and an additional 5267 were identified in the update search in 2018. Overall, 61 titles and abstracts were considered potentially relevant and full papers were retrieved and screened. Our original review included five studies. We identified one additional relevant study through our updated searches. A total of six studies (total 341 patients, sample size range 15–100) were therefore included in the review of diagnostic accuracy (figure 1). Where reported, the mean age of patients was in the mid-forties, the majority were male, and most had had symptoms for at least 3 months. One study excluded patients with a previous history of lumbar surgery,[27] in contrast a substantial minority of patients (up to 48%) had had previous surgery in two of the other studies. Details of the patients included, and the injections delivered in each study are given in table 1 (online supplementary table 1).

Four diagnostic cohort studies (one prospective and three retrospective) recruited patients with suspected LR in whom some doubt remained due to equivocal or discordant clinical and radiological findings. Schutz *et al* and Dooley *et al* used intraoperative findings during surgery as the reference standard (table 2).[28 29] In addition, Dooley *et al* used outcome following surgery as a second reference standard.[29] Williams and Germon and Sasso *et al* used outcome following surgery at 3 and 12 months,[30 31] respectively, as the reference standard.

Two studies had a within-patient case-control study design. In the Yeom *et al* study, control injections were given at adjacent asymptomatic nerve roots,[27] whereas in the North *et al* study, other anatomic sites in the lumbar spine were injected (sciatic nerve, facet joint and subcutaneous).[32] All cases were confirmed by concordant clinical and radiological or surgical findings prior to the use of SNRB.

### Quality of included studies

All studies were judged at high risk of bias (table 3). All studies had high risk of bias for the reference standard because postsurgical outcomes were not considered[27 32] or selectively measured[28–31] (eg, surgery was predominantly

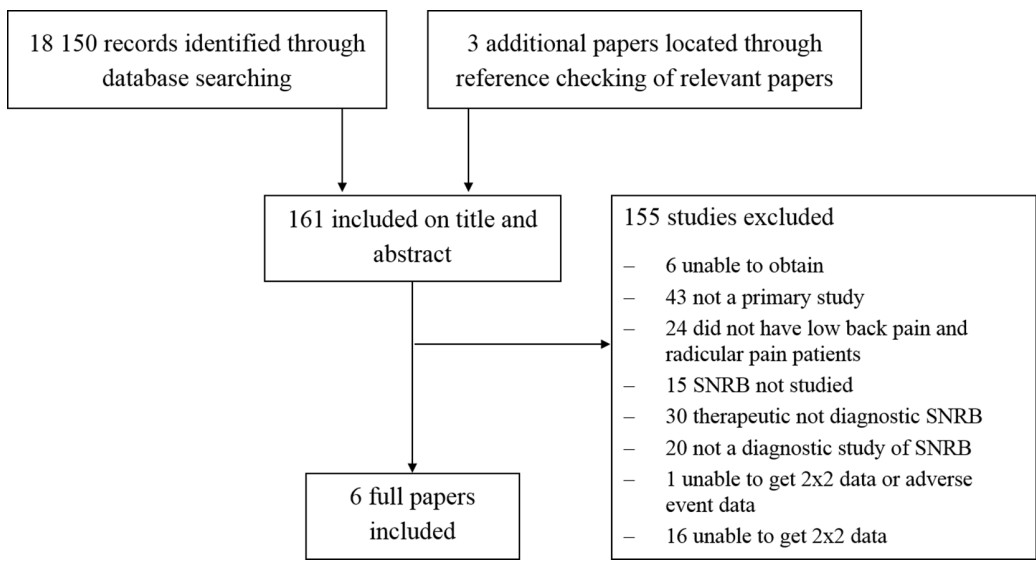

**Figure 1** Flowchart of diagnostic study selection process. SNRB, selective nerve root block.

The boxes in the flowchart read:

- 18 150 records identified through database searching
- 3 additional papers located through reference checking of relevant papers
- 161 included on title and abstract
- 155 studies excluded
  - 6 unable to obtain
  - 43 not a primary study
  - 24 did not have low back pain and radicular pain patients
  - 15 SNRB not studied
  - 30 therapeutic not diagnostic SNRB
  - 20 not a diagnostic study of SNRB
  - 1 unable to get 2x2 data or adverse event data
  - 16 unable to get 2x2 data
- 6 full papers included

**Table 1** Details of included studies

| Author (year) country | N analysed/N recruited | Description of included patients | Details of previous surgery | Needle level | Anaesthetic details | Guided method | Needle provocation | Number of control injections | Time to pain measurement |
|---|---|---|---|---|---|---|---|---|---|
| **Within-patient case–control studies** | | | | | | | | | |
| Yeom et al[27] (2008) NR | 47/83 | Established single-level radiculopathy. Concordant imaging and clinical findings | No history of lumbar surgeries | L3, L4, L5, S1 | 1 mL of 2% lidocaine | Fluoroscopy | No | 1 or 2 | 30 min |
| North et al[32] (1996) USA | 33/33 | Established sciatica with or without low back pain. History of nerve root compression or imaging findings of ongoing nerve root compression | 48% history of root compression corrected surgically | L5, S1 | 3 mL of 0.5% bupivacaine | Fluoroscopy | Yes | 3 | Every 15 min for 3 hours |
| **Prospective diagnostic cohort studies** | | | | | | | | | |
| Schutz et al[28] (1973) Canada | 15/23 | Current sciatica symptoms | Unclear if patients included in analysis had previous surgeries | NR | 1 mL of procaine (concentration NR) | Guided but method NR | Yes | 1 or 2 | Immediate |
| **Retrospective diagnostic cohort studies** | | | | | | | | | |
| Sasso et al[31] (2005) USA | 83/83 | Cervical or lumbar radiculopathy. Discordant imaging and clinical findings | Unclear how many previous lumbar surgeries | NR | 0.5–0.7 mL of 2% lidocaine | Fluoroscopy | Yes | NR | Immediate |
| Dooley et al[29] (1988) Canada | 62/73 | Radicular pain and previous nerve root infiltration | 32≥1 previous surgery, 3 had 4 surgeries | L3, L4, L5, S1 | 1 mL of 1% mepivacaine or lidocaine | Fluoroscopy | Yes | NR | Immediate |
| Williams et al[30] (2015) UK | 96/100 | Presumed radicular leg pain. Discordant clinical and imaging findings | NR | L1, L3, L4, L5, S1 | 2 mL of 1% lidocaine and 0.5–1 mL of Iopamidol | Fluoroscopy | Yes | NR | Immediate |

DRGB, dorsal root ganglion block; NR, not reported; SNRB, selective nerve root block.

**Table 2** Diagnostic accuracy results

| Author (year) | Threshold | Reference standard | TP | FN | Sensitivity % (95% CI) | TN | FP | Specificity % (95% CI) | PLR (95% CI) | NLR (95% CI) |
|---|---|---|---|---|---|---|---|---|---|---|
| Within-patient case–control studies* | | | | | | | | | | |
| Yeom et al[27] (2008) | 70% pain relief—several other thresholds also evaluated | Concordant symptoms and imaging evidence of compression (case injections) or no symptoms or imaging evidence of compression (control injections) | 27 | 20 | 57 (43 to 70) | 50 | 8 | 86 (75 to 93) | 4.1 (2.1 to 8.3) | 0.5 (0.4 to 0.7) |
| North et al[32] (1996) | 50% reduction in baseline pain following block | Concordant symptoms and imaging evidence of compression (case injections) or no symptoms or imaging evidence of compression (control injections) | 30 | 3 | 91 (76 to 97) | 8 | 25 | 24 (12 to 41) | 1.2 (1.0 to 1.5) | 0.4 (0.1 to 1.3) |
| Diagnostic cohort studies | | | | | | | | | | |
| Schutz et al[28] (1973) | 100% pain relief. Full trunk flexion and straight leg raising possible | Intraoperative findings | 12 | 0 | 100 (76 to 100) | 1 | 2 | 33 (6 to 79) | 1.5 (0.7 to 3.3) | 0.0 |
| Sasso et al[31] (2005) | Visual Analogue Scale score 0–1 and immediate relief of >95% pain | Outcome 12 months following surgery | 71 | 3 | 96 (89 to 99) | 5 | 4 | 56 (27 to 81) | 2.2 (1.0 to 4.5) | 0.1 (0.0 to 0.3) |
| Dooley et al[29] (1988) | Pain relief | Intraoperative surgical confirmation of root pathology | 46 | 4 | 92 (81 to 98) | 2 | 1 | 67 (9 to 99) | 2.8 (0.6 to 13.7) | 0.1 (0.0 to 0.4) |
| | Pain relief | Outcome following surgery (follow-up range 24–36 months) | 28 | 4 | 88 (71 to 96) | 2 | 19 | 10 (1 to 30) | 1.0 (0.8 to 1.2) | 1.2 (0.3 to 6.5) |
| Williams et al[30] (2015) | Pain relief | Outcome 3 months following surgery (resolution of symptoms) | 41 | 7 | 85 (72 to 94) | 2 | 10 | 17 (3 to 48) | 1.0 (0.8 to 1.4) | 0.9 (0.2 to 3.7) |

*Please note that the unit used in the within-patient case–control studies is number of injections and some patients had two control injections at adjacent levels in addition to the affected nerve; in all other studies it is number of patients.

FN, false negative; FP, false positive; NLR, negative likelihood ratio; PLR, positive likelihood ratio; TN, true negative; TP, true positive.

**Table 3** QUADAS-2 results

| Author (year) | Risk of bias | | | | Applicability concerns | | |
|---|---|---|---|---|---|---|---|
| | Patient selection | Index test | Reference standard | Flow and timing | Patient selection | Index test | Reference standard |
| **Within-patient case-control studies** | | | | | | | |
| Yeom et al[27] (2008) | 🙁 | 🙁 | 🙁 | 🙁 | 🙁 | 🙂 | 🙁 |
| North et al[32] (1996) | 🙁 | 🙂 | 🙁 | 🙂 | 🙁 | 🙂 | 🙁 |
| **Diagnostic cohort studies** | | | | | | | |
| Sasso et al[31] (2005) | 🙂 | 🙂 | 🙁 | 🙁 | 🙂 | 🙂 | 🙂 |
| Schutz et al[28] (1973) | ? | 🙂 | 🙁 | 🙁 | ? | 🙂 | 🙁 |
| Dooley et al[29] (1988) | 🙂 | 🙂 | 🙁 | 🙁 | 🙂 | 🙂 | 🙂 |
| Williams et al[30] (2015) | 🙂 | ? | 🙁 | 🙁 | 🙂 | 🙂 | 🙂 |

🙁, low risk/concern; 🙂, high risk/concern; ?, unclear risk/concern.

performed in patients with positive SNRB findings). The four cohort studies were at high risk of bias for flow and timing because patients were selected to undergo surgery based on the SNRB result, with patients testing positive more likely to receive surgery. It is likely that the patients with negative SNRB results who, despite this, were selected for surgery were a biased subset of those testing negative as these are likely to have been the patients in whom the clinicians suspected a false negative result. The two within-patient case-control studies were at high risk of bias and poor applicability for patient selection because they recruited patients with unequivocal and concordant imaging and clinical findings rather than patients where diagnostic uncertainty remained. Three cohort studies were judged as low concerns regarding applicability on all domains.[29–31] There were high concerns regarding the applicability of the fourth cohort study as the reference standard consisted of intraoperative findings alone.[28]

### Summary of test accuracy results

The diagnostic cohort studies reported data at the patient level, but only data at the injection level were available for the within-patient case-control studies. The threshold used to determine a positive SNRB test varied between studies (table 2). We decided not to pool the results of studies that used different reference standards.

There was substantial heterogeneity in estimates of sensitivity and specificity across studies; sensitivity ranged from 57% to 100% and specificity from 10% to 86% (table 2, figure 2). Sensitivity exceeded 85% in all studies except Yeom et al (57%).[27] Specificity was lower than 75% in all studies except Yeom et al (86%).[27] Interpretation of specificity is particularly hampered by verification bias in the cohort studies. Because surgeons were not blinded to the SNRB results, very few patients with negative test findings had surgery. Williams and Germon,

Sasso et al, Schutz et al and Dooley et al contributed a total of just 10 true negative cases.[28–31] The higher specificity reported by Yeom et al could be a manifestation of patient selection bias as 'control' injections were performed at a level of the spine where the patients had no symptoms or imaging findings suggestive of pathology.[27] Positive likelihood ratios were generally low (<5), meaning that a positive SNRB result did not greatly increase the posttest probability that the nerve root was the source of the low back and radicular pain.

Due to the patient selection bias inherent in within-patient case-control designs, we decided that it would be inappropriate statistically to combine their results with those of the diagnostic cohort studies, and because of differences in the type of control injection we did not pool the results of the two studies. Based on the two

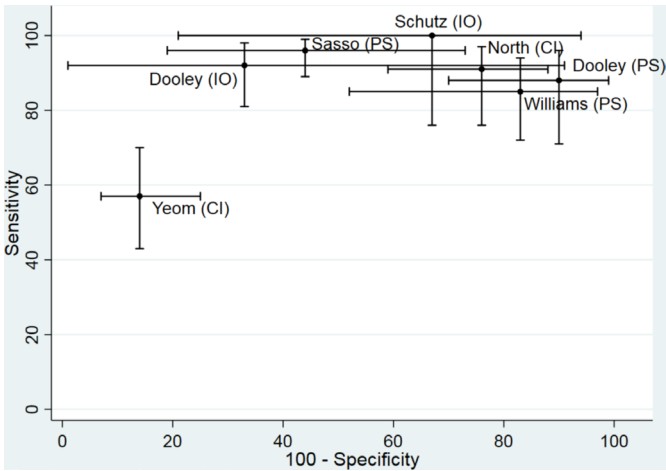

**Figure 2** ROC plot displaying diagnostic accuracy results of included studies. CI, control injection reference standard; IO, intraoperative reference standard; PS, postsurgical reference standard; ROC, receiver-operating characteristic.

cohort studies that used an intraoperative reference standard the pooled sensitivity was 93.5% (95% CI 84.0% to 97.6%) and specificity was 50.0% (16.8% to 83.2%). For the three studies that used postsurgery as the reference standard, the summary sensitivity was 90.9% (83.1% to 95.3%) and summary specificity was 22.0% (7.4% to 49.9%). Low specificity implies that a high proportion of patients without nerve root compromise have a positive SNRB result.

## Adverse events review

Eight studies assessed complications and/or adverse events (online supplementary table 2).[28 30 33–38] Two were diagnostic cohorts,[28 30] one was a randomised controlled trial[34] and five were case series.[33 35–38] Only one reported the complications of SNRBs in the lumbar spine as the primary outcome.[33] Five studies reported that there were no complications. Tajima *et al* reported aggravated pain in the lower extremity for 1–2 days following selective radiculography and block in 4 (3.8%) patients.[38] The largest study reported that minor and transient complications were encountered in 98 of the 1777 total patient visits (during which 2217 injections were delivered to 1203 patients), giving an overall per patient visit complication rate of 5.5%.[33] Complications occurred in 134 of the 2217 total injections (6% complication rate per injection). There were no major or permanent complications resulting from SNRB in this large case series.

## DISCUSSION

Despite the long-standing use of SNRB to help in the selection of patients who might benefit from surgery and in guiding the surgical approach, few studies have estimated its diagnostic accuracy. Our systematic review identified six studies, all at high risk of bias. Many were at risk of verification bias, because patients with positive SNRB were more likely to undergo surgery than those testing negative. There was substantial variation in estimates of sensitivity and specificity across studies. Based on the three cohort studies that used postsurgery outcomes as the reference standard, the summary sensitivity was 90.9% (83.1% to 95.3%) and summary specificity was 22.0% (7.4% to 49.9%). SNRB is a safe test with a low risk of clinically significant complications, but it remains unclear whether the additional diagnostic information it provides, improves patient outcomes or justifies the cost of the test.

Extensive literature searches were conducted in an attempt to locate all relevant studies. These included electronic searches in a wide variety of databases, scanning the references of included studies and previous systematic reviews. Diagnostic accuracy studies are difficult to identify from electronic databases as there are no specific indexing terms. Therefore, very sensitive searches were carried out to ensure that relevant studies were not missed. It is unlikely that any relevant published studies have been missed, although it is possible that some unpublished studies were not identified. The small number of primary diagnostic accuracy studies included in our review, all had methodological limitations. Due to the small number of studies, we were unable to explore the value of SNRB in potentially important patient subgroups, such as those with suspected multilevel radiculopathy.

Four previous systematic reviews of the diagnostic utility of SNRB in patients whose pain was of spinal origin have been reported.[15–18] The two earlier reviews had positive interpretations of the data and concluded that there was moderate evidence for SNRB in the 'pre-operative evaluation of patients with negative or inconclusive imaging studies, but with clinical findings of nerve root irritation'.[16 18] More recent reviews, however, concluded that there was limited evidence for the accuracy of SNRB as a diagnostic tool.[15 17] Our update review shows similar results. We found limited evidence which was of low methodological quality indicating that the diagnostic accuracy of SNRB is uncertain and that specificity in particular may be low. The differences in interpretation between our review and those conducted previously may be partly due to the smaller number of primary studies included in our review. We used rigorous eligibility criteria, which excluded studies with mixed cervical and lumbar spine pathology and studies where there was insufficient data to construct estimates of sensitivity and specificity.

For centres that currently rely on SNRB for diagnostic information to help decide whether, or at which level, to perform lumbar decompressive surgery, it is vital that better evidence is generated. Moreover, according to Hospital Episodes Statistics (HES), which contains records of all admissions, appointments and attendances for patients at NHS hospitals in England, 58 399 injections of therapeutic substance around spinal nerve root took place from 1 April 2016 to 31 March 2017.[8] Due to the granularity of HES data, it is not possible to tell how many of these injections were diagnostic lumbar SNRB. Nevertheless, the number is substantial, and it is therefore apparent that the community of spinal surgeons has a responsibility to generate robust evidence for the use of diagnostic SNRBs. A methodologically ideal diagnostic accuracy study is unlikely to be clinically acceptable as it would require all patients, including those with negative SNRB findings, to undergo surgery. Furthermore, while diagnostic accuracy studies can explore whether SNRB accurately predicts surgical outcomes, they cannot answer the more fundamental question of whether SNRB improves surgical decisions and patient outcomes. Much better evidence would be provided by a trial randomising patients who are being considered for surgery but have discordant or equivocal clinical and imaging findings of nerve root compression to receive a diagnostic SNRB or to have management based on clinical and imaging findings alone. Given the lack of high quality evidence on the diagnostic accuracy of SNRB, we believe that such a trial would be ethically acceptable and would help patients, clinicians and healthcare payers decide whether SNRB

can improve patient outcomes by targeting surgery at those most likely to benefit.

Finally, it should be mentioned that this systematic review did not consider the use of SNRBs as a therapeutic option for patients with radicular pain due to a prolapsed lumbar intervertebral disc. The most recent National Institute for Heath and Care Excellence (NICE) guidance concluded that the evidence for both image guided and non-image guided injections for patients with acute and severe sciatica was mostly low or moderate.[8] However, the guidance recommends that an injection of local anaesthetic and steroid should be considered in acute, severe sciatica where patients would otherwise be offered surgery. The NErve Root Block VErsus Surgery (NERVES) randomised trial, which enrolled patients in 12 NHS hospitals, aimed to compare surgical microdiscectomy versus SNRB in patients with sciatica of at least 6 weeks' duration secondary to a prolapsed intervertebral disc. The results of this trial, which is currently in follow-up, will elucidate the role of SNRB as a therapeutic but not diagnostic option. Hence, it is important that consideration is given to a trial of diagnostic SNRB as outlined above.

## CONCLUSIONS

There is no high-quality evidence on the diagnostic accuracy of SNRB in patients with radiculopathy and discordant or equivocal imaging findings. The evidence that is available suggests that the specificity of SNRB is low. As there is no adequate reference standard for determining the diagnostic accuracy of SNRB, future research should focus on randomised controlled trials to evaluate whether SNRB improves the process of care or patient outcomes.

**Author affiliations**
[1]Population Health Sciences, Bristol Medical School, University of Bristol, Bristol, UK
[2]The National Institute for Health Research Collaboration for Leadership in Applied Health Research and Care West (NIHR CLAHRC West), University Hospitals Bristol NHS Foundation Trust, Bristol, UK
[3]Radiology, Addenbrooke's Hospital, Cambridge, UK
[4]Division of Neurosurgery, Department of Clinical Neurosciences, Addenbrooke's Hospital & University of Cambridge, Cambridge, UK

**Acknowledgements** The authors wish to thank Margaret Burke for her advice in developing and implementing the search Strategy, Catherine Jameson for implementing the original searches and reviews and Alison Richards for conducting the update searches. This research was supported by the National Institute for Health Research (NIHR) Collaboration for Leadership in Applied Health Research and Care West (NIHR CLAHRC West) and the NIHR Health Technology Assessment programme. The views expressed in this article are those of the author(s) and not necessarily those of the NHS, the NIHR, or the Department of Health and Social Care.

**Contributors** RB conducted the reviews of diagnostic accuracy and adverse events, conducted analyses and completed the first draft of the manuscript. MMCE updated the review, including abstract and full text screening, data extraction, risk of bias assessment, analysis and updated the draft of the manuscript. AS updated the review, including abstract and full text screening and checking of data extraction and risk of bias assessment. RJL contributed to the conception and design of the study, provided clinical expertise in neurosurgery, helped with acquisition of data for the service evaluation of SNRB and critically revised the manuscript. AGK provided clinical expertise in neurosurgery and critically revised the manuscript. JNH was involved in the inception of study and critical appraisal

of literature, particularly from a radiological perspective. PW contributed to the conception and design of the study, supervised conduct of reviews of diagnostic accuracy and adverse events and critically revised the manuscript. WH was principal investigator on the project, contributed to the conception and design of the study, supervised conduct of reviews of diagnostic accuracy and adverse events and critically revised the manuscript. All authors approved the manuscript for publication.

**Funding** PW and MMCE are funded by a National Institute for Health Research Collaboration for Leadership in Applied Health Research and Care West (NIHR CLAHRC West). AS is funded by an NIHR Systematic Review Fellowship (RM-SR-2017-08-012). AGK is supported by a Clinical Lectureship, School of Clinical Medicine, University of Cambridge. JNH, RB, RJL, PW and WH were funded by a National Institute for Health Research Health Technology Assessment programme grant (project number 09/111/01).

**Competing interests** None declared.

**Patient consent for publication** Not required.

**Provenance and peer review** Not commissioned; externally peer reviewed.

**Data sharing statement** There is no identifiable patient data included in the manuscript. All included data are extracted from published trials and references are included.

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
