## [Reviewer comments · BMJ Open]

ARTICLE DETAILS

TITLE (PROVISIONAL)	The utility of diagnostic selective nerve root blocks in the management of patients with lumbar radiculopathy: a systematic review
AUTHORS	Beynon, Rebecca; Elwenspoek, Martha; Sheppard, Athena; Higgins, John; Koliass, Angelos; Laing, Rodney; Whiting, Penny; Hollingworth, William

VERSION 1 - REVIEW

REVIEWER	Laxmaiah Manchikanti Pain Management Center of Paducah Paducah, KY US
REVIEW RETURNED	16-Aug-2018

GENERAL COMMENTS	Authors have described the utility of diagnostic selected nerve root blocks in the management of patients with lumbar radiculopathy. This is a systematic review without metaanalysis. Overall, subject is of interest; however, there have not been any recent studies validating these procedures. There have been multiple systematic reviews performed in the past. Authors are essentially using the same studies. They have performed a systematic review in recent past. The manuscript is well conceived and well written. Authors as noted have performed similar review along with cost effectiveness of selective nerve root blocks as quoted in the manuscript. This present review identified only one study; thus, it reduces the importance of this study. Authors have performed appropriate quality assessment with risk of bias; however, authors have not performed a qualitative review or quantitative review. It may be appropriate to perform qualitative review, at least with best evidence synthesis with grading of evidence. Authors may also considering performing a single-arm metaanalysis if it is feasible.
--

REVIEWER	chris lavy Nuffield Orthopaedic Centre, Oxford University, UK
REVIEW RETURNED	27-Aug-2018

GENERAL COMMENTS	This is a well thought out and well executed literature review. It starts with the important clinical question of whether elective nerve root blocks are accurate in identifying patients likely to benefit from surgery. The review is carefully done but perhaps unsurprisingly there is no clear evidence of diagnostic accuracy. In my opinion it is worth publishing as almost all practicing spine surgeons will be involved in SNRBs or at least discussing them in MDTs. I do not think that there needs to be any major change in the paper but I have two comments below. 1 This is a slightly pedantic point but on line 5 in the abstract and in some other places in the text it is stated that "lumbar radiculopathy causes low back pain accompanied by pain radiating to the legs." In a small number of cases there is no back pain and often the leg pain is unilateral, thus it would in my opinion be safer to make those points a bit clearer. 2 The statistical tests appear appropriate to me, but I feel that they should be reviewed by someone with more expertise in that field.
--

REVIEWER	Kukuh Noertjojo Evidence-Based Practice Group, Clinical Services, WorkSafeBC. CANADA.
REVIEW RETURNED	22-Oct-2018

GENERAL COMMENTS	It has been my pleasure to review your manuscript. This is not only an important topic that you reviewed (including in my line of work), but I also commended you on the robustness of your methodology and the clarity of your presentation. Considering the importance of this topic, I am hoping that the Editor will fast track your publication and NICE will include this finding on her next update on low back pain guidelines. Aside of this, I'd like to ask you if it is possible for you to add/discuss about:  - Positive predictive value - Negative predictive value - Likelihood ratio of positive and negative test - Post-test likelihood of positive and negative test on top of sensitivity and specificity numbers that you've presented. Also on Table 3. Can you make the smiley faces in color and bigger? it will be easier to glance/read. Thank you for letting me review your excellent manuscript.
---

VERSION 1 – AUTHOR RESPONSE

Reviewer: 1

Reviewer Name: Laxmaiah Manchikanti

Institution and Country: Pain Management Center of Paducah, Paducah, KY, US

Authors have described the utility of diagnostic selected nerve root blocks in the management of patients with lumbar radiculopathy. This is a systematic review without metaanalysis.

Overall, subject is of interest; however, there have not been any recent studies validating these procedures. There have been multiple systematic reviews performed in the past. Authors are essentially using the same studies. They have performed a systematic review in recent past.

The manuscript is well conceived and well written. Authors as noted have performed similar review along with cost effectiveness of selective nerve root blocks as quoted in the manuscript. This present review identified only one study; thus, it reduces the importance of this study.

Authors have performed appropriate quality assessment with risk of bias; however, authors have not performed a qualitative review or quantitative review. It may be appropriate to perform qualitative review, at least with best evidence synthesis with grading of evidence.

Authors may also consider performing a single-arm metaanalysis if it is feasible.

Answer:

Contrary to the reviewer's comment, we did meta-analyse the findings of the two cohort studies that used an intra-operative reference standard and the three cohort studies that used post-surgery outcomes as the reference standard. We found that SNRB had high sensitivity but low specificity and report the results on page 11. We do not think a single-arm meta-analysis or additional qualitative review would add value over and above the meta-analyses that we have done.

SNRB is widely used as a diagnostic tool despite the very limited evidence, identified in our paper, supporting its diagnostic accuracy. Our paper is important and relevant for clinicians and policy makers as it challenges them to reconsider current policies on the use of SNRB and to develop better evidence to inform future policy.

Reviewer: 2

Reviewer Name: chris lavy

Institution and Country: Nuffield Orthopaedic Centre, Oxford University, UK

This is a well thought out and well executed literature review. It starts with the important clinical question of whether elective nerve root blocks are accurate in identifying patients likely to benefit from surgery. The review is carefully done but perhaps unsurprisingly there is no clear evidence of diagnostic accuracy. In my opinion it is worth publishing as almost all practicing spine surgeons will be involved in SNRBs or at least discussing them in MDTs. I do not think that there needs to be any major change in the paper but I have two comments below.

1 This is a slightly pedantic point but on line 5 in the abstract and in some other places in the text it is stated that "lumbar radiculopathy causes low back pain accompanied by pain radiating to the legs." In a small number of cases there is no back pain and often the leg pain is unilateral, thus it would in my opinion be safer to make those points a bit clearer.

2 The statistical tests appear appropriate to me, but I feel that they should be reviewed by someone with more expertise in that field.

Answer:

1) We have clarified these explanations in the manuscript as follows:

a. Abstract, page 2, line 2: "Lumbar radiculopathy often manifests as pain in the lower back radiating into one leg (sciatica)."

b. Abstract, page 2, line 8: "Primary research articles using a patient population with low back pain and symptoms in the leg, [...]"

c. Introduction page 3, line 3-4: "In a subgroup of patients, low back pain is accompanied by pain radiating to a lower extremity in a radicular distribution (sciatic pain). Leg pain is one of the symptoms of lumbar radiculopathy (LR) but other symptoms, such as numbness, tingling, weakness, can also develop."

2) No response from authors required.

Reviewer: 3

Reviewer Name: Kukuh Noertjojo

Institution and Country: Evidence-Based Practice Group, Clinical Services, WorkSafeBC. CANADA.

It has been my pleasure to review your manuscript. This is not only an important topic that you reviewed (including in my line of work), but I also commended you on the robustness of your methodology and the clarity of your presentation. Considering the importance of this topic, I am hoping that the Editor will fast track your publication and NICE will include this finding on her next update on low back pain guidelines.

Aside of this, I'd like to ask you if it is possible for you to add/discuss about:

- Positive predictive value
- Negative predictive value
- Likelihood ratio of positive and negative test
- Post-test likelihood of positive and negative test

on top of sensitivity and specificity numbers that you've presented.

Also on Table 3. Can you make the smiley faces in color and bigger? it will be easier to glance/read.

Thank you for letting me review your excellent manuscript.

Answer:

- We have added the likelihood ratios of positive and negative tests in Table 2 in the manuscript and we have adapted the results accordingly:

- o Page 11, line 19-20: "PLR was generally low (<5), meaning that a positive SNRB result did not greatly increase the post-test probability that the nerve root was the source of the low back and radicular pain."

- We did not add PPV/NPV or positive/negative post-test odds as we want to keep the message of the paper focussed on diagnostic accuracy and these statistics reflect both accuracy and prevalence (which varied between studies), blurring the message.

- We have increased the font size of the smileys and added color coding to Table 3.

VERSION 2 – REVIEW

REVIEWER	Laxmaiah Manchikanti, MD. Pain Management Center of Paducah Paducah, KY US
REVIEW RETURNED	16-Jan-2019

GENERAL COMMENTS	Authors have described the utility of diagnostic selected nerve root blocks in the management of patients with lumbar radiculopathy. This is a systematic review without metaanalysis. Overall, subject is of interest; however, there have not been any recent studies validating these procedures. There have been multiple systematic reviews performed in the past. Authors are essentially using the same studies. They have performed a systematic review in recent past. The manuscript is well conceived and well written. Authors as noted have performed similar review along with cost effectiveness of selective nerve root blocks as quoted in the manuscript. This present review identified only one study; thus, it reduces the importance of this study. Authors have performed appropriate quality assessment with risk of bias; however, authors have not performed a qualitative review or quantitative review. It may be appropriate to perform qualitative review, at least with best evidence synthesis with grading of evidence. Authors may also considering performing a single-arm metaanalysis if it is feasible. There is no meta-analysis available.
--

REVIEWER	chris lavy Nuffield departement of orthopaedics rheumatology and musculoskeletal science, Oxford England
REVIEW RETURNED	09-Jan-2019

GENERAL COMMENTS	I am happy that the issues identified in the first review have been addressed
---

REVIEWER	Kukuh Noertjojo Evidence-Based Practice Group, Clinical Services, WorkSafeBC. CANADA
REVIEW RETURNED	08-Jan-2019

GENERAL COMMENTS	I thank you for the revision you have done. As I mentioned in my previous review, this systematic review is timely and I am hoping that it will help changing practice.
---